# Immune Checkpoint Inhibitor Associated Myocarditis and Cardiomyopathy: A Translational Review

**DOI:** 10.3390/biology12030472

**Published:** 2023-03-20

**Authors:** Dong Wang, Johann Bauersachs, Dominik Berliner

**Affiliations:** Department of Cardiology and Angiology, Hannover Medical School, 30625 Hannover, Germany

**Keywords:** immune checkpoint inhibitor, cardiomyopathy, heart failure, myocarditis, adverse effects, cardio-oncology

## Abstract

**Simple Summary:**

Immune checkpoint inhibitors (ICIs) are novel cancer therapeutics that address the immune system to attack malignancies. While this therapy has significantly improved the outcome of a variety of tumor patients, it has been associated with a new class of side effects, so-called immune-related adverse effects (irAEs). IrAEs can appear in various forms and involve any organ. Cardiovascular irAEs are highly feared due to their very high mortality and morbidity, and their underlying pathomechanisms are far from being understood. Researchers in immunology, molecular biology, and translational medicine need to work hand in hand to close these gaps in knowledge. At the same time, it is crucial for healthcare professionals, particularly cardiologists and oncologists, to have a good understanding of this increasingly common clinical problem. In this review, we discuss the various forms of cardiotoxicity of ICI therapy with emphasis on the currently known pathophysiology, clinical presentation, and diagnostics, as well as available treatment strategies.

**Abstract:**

Immune checkpoint inhibitors (ICIs) have revolutionized oncology and transformed the treatment of various malignancies. By unleashing the natural immunological brake of the immune system, ICIs were initially considered an effective, gentle therapy with few side effects. However, accumulated clinical knowledge reveals that ICIs are associated with inflammation and tissue damage in multiple organs, leading to immune-related adverse effects (irAEs). Most irAEs involve the skin and gastrointestinal tract; however, cardiovascular involvement is associated with very high mortality rates, and its underlying pathomechanisms are poorly understood. Ranging from acute myocarditis to chronic cardiomyopathies, ICI-induced cardiotoxicity can present in various forms and entities. Revealing the inciting factors, understanding the pathogenesis, and identifying effective treatment strategies are needed to improve the care of tumor patients and our understanding of the immune and cardiovascular systems.

## 1. Introduction

In the last decade, major advancements in immune therapy, specifically immune checkpoint inhibitors (ICIs), have ameliorated the treatment of an increasing number of oncological patients. ICIs are monoclonal antibodies that target inhibitory immune receptors expressed on T-lymphocytes, antigen-presenting cells, and tumor cells. They elicit an anti-tumor response by stimulating the immune system. 

Initially approved by the FDA for metastatic malignant melanoma, the indication for ICI therapy has expanded today to more than 85 malignancies [1] and continues to extend. Currently, more than 3000 ICI trials are ongoing, representing about 2/3 of all oncology trials [2]. More importantly, ICIs are now being utilized in earlier disease stages and are used as single agents or in combination with chemotherapies as first- or second-line treatments [3]. About 50% of cancer patients are eligible for ICI treatment [4,5]. 

While ICI therapy has improved the outcomes of a variety of tumors, new, radically different side effects from previous treatments, cytotoxic or targeted therapies, have appeared. Given that ICIs act to unleash the immune system, a broad spectrum of sometimes ill-defined adverse events, referred to as immune-related adverse events (irAEs), may manifest in a variety of organs with severe or even fatal results. Besides colitis, dermatitis, thyreoiditis, pneumonitis, hepatitis, and hypophysitis, cardiovascular irAEs have emerged as infrequent complications [6,7,8] and can manifest as arrhythmias, myocarditis, pericarditis, vasculitis, cardiomyopathy, and possibly atherosclerosis [9,10]. 

Its sporadical appearance may reflect a failure to recognize mild presentations and our insufficient understanding of its manifestation. Furthermore, the precise mechanisms by which ICIs cause cardiovascular irAEs remain unknown. 

In this review, we provide an overview of cardiac aspects of ICI therapy with a focus on its clinical manifestation, diagnostic, and treatment approaches. 

## 2. Molecular Mechanisms of Immune Checkpoints

A major challenge of the immune system is to differentiate between harmful microbial or oncological targets and self-antigens or foreign antigens that are non-dangerous (such as symbiotic bacteria or paternal antigens in the fetus). One major mechanism of immune tolerance is the expression of coinhibitory receptors on the surface of T cells, referred to as immune checkpoints. 

T-cells are classified into CD4^+^ helper and CD8+ cytotoxic cells. As the name implies, CD4^+^ helper cells promote immune reactions by inducing B-cell antibody class switching, breaking cross-tolerance in dendritic cells, activating CD8+ cytotoxic T-cells, and maximizing the bactericidal activity of phagocytes such as macrophages and neutrophils [11]. In contrast, CD8+ cytotoxic cells act directly by killing target cells [12]. 

For a T-cell to become activated, it requires the presentation of antigens on MHC I/II molecules to the T-cell receptor (TCR) by antigen-presenting-cells (APCs) such as dendritic cells and a second activating signal (also known as co-stimulation). Besides co-stimulatory signals, T-cell activation is also modulated by co-inhibitory signals. The summation of co-regulatory signals (co-stimulatory and co-inhibitory) drives T-cell activation or anergy. 

The best-studied costimulatory ligands are CD80 (B7-1) and CD86 (B7-2), which are expressed on APCs and bind to CD28 on T-cells [13,14]. They belong to the immunoglobulin superfamily. Upon contact with infected or diseased cells, APCs upregulate CD80 and CD86, which bind to CD28 on T-cells and thereby drive them into activation [15,16]. CD28 binding activates phosphoinositid-3-kinase (PI3K), growth factor receptor-bound protein 2 (GRB2), GRB2-related adapter protein 2 (GADS), protein kinase C theta (PKCθ) and Lck, inducing the NFkB, NFAT, and AP-1 pathways. They eventually lead to cell proliferation, reduced apoptosis, and the secretion of IL-2 [17]. To prevent uncontrolled T-cell activation, co-inhibitory receptors are expressed on T cells. Cytotoxic T lymphocyte–associated protein 4 (CTLA-4; CD152) is highly homologous to CD28 and competes for the same ligands as CD80 and CD86 [18]. CTLA-4 has opposing effects to CD28 and inhibits T-cell activation by inhibiting the PI3K/AKT/mTOR pathway [19,20]. It binds to CD86 and CD80 with a higher affinity than CD28 [21]. Thus, depending on the relative abundance of CTLA-4 and CD28, CTLA-4 can outcompete CD28 for CD80 and CD86 and inhibit T-cell activation [22]. In CD4^+^ T cells, this leads to differentiation into inhibitory T-regulatory cells (Tregs) rather than excitatory Th1-cells [23]. In addition, CTLA-4 is constitutively expressed on Tregs, contributing to their ability to suppress immune activation and tolerance [24]. In CD8+ T-cells, the activation of CTLA-4 inhibits their function and leads to T-cell exhaustion, which is characterized by reduced cytokine production and cytotoxic activity [23].

A second well-studied co-regulatory receptor is programmed cell death protein 1 (PD-1 or CD279) and its cognate ligands PD-L1 and PD-L2. PD-1 is expressed on activated T-cells, B-cells, NK-cells, monocytes, dendritic cells, and constitutively on Tregs [25]. While PD-L2 expression is limited to macrophages and dendritic cells, PD-L1 is expressed by the majority of hematopoietic cells as well as a number of non-hematopoietic cells, like endothelial cells or cardiomyocytes [26,27]. When PD-1 binds to its ligands, it leads to intracellular recruitment of SHP1 and SHP2 (Src homology 2 domain-containing protein tyrosine phosphatases) to dephosphorylate and inactivate ZAP7 and downregulation of the PI3K/AKT/mTOR pathway, which eventually dampens T-cell activation, migration, and proliferation [28,29]. Both CD4 and CD8+ T cell subsets are susceptible to this inhibitory pathway [30,31]. Hence, in contrast to CTLA-4, which inhibits mainly by outcompeting CD86 and CD86 ligands on APCs away from CD28, PD-1 inhibits T-cell activation through a cell-intrinsic signaling mechanism. In addition, the constitutive expression of PD-L1 in Tregs contributes significantly to the immunosuppressive function of Tregs [32].

Besides the above-mentioned receptors, many other receptors are expressed on T-cells and hence regulate cell activation. Co-stimulatory signals can be differentiated into two major classes: the immunoglobulin superfamily, which consists of CD28 and inducible T-cell costimulatory (ICOS), and the tumor necrosis factor (TNF) superfamily [15,17,33,34,35,36,37]. CD27, CD40, OX40, GITR, and 4-1BB can bind to TNF and belong to the TNF superfamily [38]. Co-inhibitory receptors include lymphocyte activation gene-3 (LAG-3), T cell immunoglobulin and mucin domain-containing-3 (TIM-3), and T-cell immunoreceptor with immunoglobulin and immunoreceptor tyrosine-based inhibitory motif domains (TIGIT) (Figure 1) [39,40,41]. While the mechanisms by which these receptors act are diverse and still unclear, their existence underscores the complexity of T-cell regulation.

## 3. Targeting Immune Checkpoints in Cancer Therapy

Cancer cells evolve various mechanisms to escape immune surveillance, such as defects in antigen presentation machinery, upregulation of negative regulatory pathways, and the recruitment of immunosuppressive cell populations [42,43,44,45,46]. One common mechanism is the overexpression of co-inhibitory receptors, for example, PD-L1, which renders cancer cells less susceptible to lysis by T-cells [47]. To counteract this mechanism, antibodies against PD-1, PD-L1, and CTLA-4 were developed over the last decade [48]. By targeting co-inhibitory receptors, these antibodies prevent activation of the co-inhibitory signal cascade, restore T-cell activation, and are referred to as immune checkpoint inhibitors (ICIs) [48]. Ipilimumab blocks CTLA-4 and was the first checkpoint inhibitor approved by the FDA in 2011 for the treatment of metastatic melanoma [49]. It has demonstrated long-term success in a significant number of patients in terms of increasing the survival rate [50]. Three years later, Nivolumab and Pembrolizumab, which block PD-1, have gained approval for the treatment of melanoma. As of today, nine ICIs have been approved by the FDA (Table 1), and their indication has expanded from initially advanced melanoma to more than eighty-five settings [1]. ICIs are used as either mono- or combination therapy and are combined with traditional chemo- and radiotherapy and target anticancer agents in both neoadjuvant and adjuvant settings [51,52,53,54,55]. Besides the well-established ICIs, which target the CTLA-4 and PD-1/PD-L1 pathways, this year the FDA granted approval for Relatlimab, the first antibody targeting LAG-3 (Table 1). New results from laboratory and clinical trials will give rise to new ICIs entering clinical practice and improve the treatment of cancer patients. However, despite the success of ICIs, immune-related adverse events are a major challenge associated with ICI treatment. 

## 4. Immune-Related Adverse Events

It was already noticed in the initial clinical trials that ICIs lead to adverse effects that are different from previously known side effects. Enhancement of immune responses by ICIs causes systemic activation of T-cell responses, leading to a range of auto-immune-like side effects, so-called irAEs. 

IrAEs generally occur in the early phase of therapy but can occur at any time, including after the termination of ICI treatment [56]. The underlying mechanism is still unclear. Different mechanisms have been postulated, ranging from the deregulation of previously tolerated self-tissue, increased cross-reactivity between cancerous and normal cells, and alteration of the humoral immunity and cytokine milieu [3,57]. Common irAEs involve the skin, gastrointestinal tract, liver, lung, and endocrine organs. Interestingly, different ICIs are associated with different irAEs. ICIs targeting CTLA-4 are more commonly associated with rash, colitis, and hypophysitis, whereas pneumonitis, hypothyroidism, arthralgia, and vitiligo are more common with PD-1/PD-L1 blockade [58,59,60]. Co-inhibition of CTLA-4 and PD-1 has been shown to increase rates of irAEs and lead to more severe complications [61]. The most fatal irAE in cases with combination therapies is myocarditis, which has a fatality rate of around 40% [62]. Initially thought to be rare complications, emerging data indicate that cardiovascular irAEs are non-negligible and are associated with a high fatality rate [63]. 

## 5. ICI-Associated Cardiotoxicity

Cardiac irAEs started to gain attention in 2016, when Johnson et al. reported two cases of fatal myocarditis following ICI treatment [64]. Since then, cardiac irAEs have expanded to include myocardial infarction, AV block, supraventricular and ventricular arrhythmias, sudden cardiac death, Tako-Tsubo cardiomyopathy, non-inflammatory cardiomyopathy, pericarditis, pericardial effusion, ischemic stroke, and venous thromboembolism [65]. In a retrospective study with 424 patients receiving at least one ICI, 62 (14.6%) patients were diagnosed with at least one new cardiovascular disease after the initiation of ICI therapy [66]. Of those, 5.6% developed heart failure under ICI monotherapy. This increased to 6.1% when two ICIs were administered sequentially [66]. Similar incidences were observed in a recent meta-analysis, which included 13,646 patients who received anti-CTLA-4, anti-PD-1, and/or anti-PD-L1 therapies. In patients receiving ICI as monotherapy, the incidence of cardiovascular adverse events was 3.1%. In patients with dual immunotherapy, the incidence nearly doubled (5.8%). Combination with chemotherapy did not majorly affect the incidence (3.7%) [67]. In a separate meta-analysis with 32,518 patients by Dolladille et al., ICI use was associated with an increased risk of myocarditis, pericardial diseases, heart failure, dyslipidemia, myocardial infarction, and cerebral arterial ischemia. Among ICI-treated patients the number needed to harm was 462 for myocarditis, whereas only 260 for heart failure [68]. Among all ICI-associated cardiovascular entities, myocarditis and non-inflammatory cardiomyopathies represent a large part and will be the focus of this review (Figure 2). A summary of clinical symptoms, diagnostic, and treatment management of ICI-associated cardiomyopathies is presented in Table 2.

### 5.1. ICI-Associated Myocarditis

#### 5.1.1. Incidence and Risk Factors

Among the various forms of cardiac irAEs, myocarditis is the most frequently reported one due to its high morbidity and mortality. As early ICI trials did not prospectively screen for myocarditis, and diagnosing myocarditis can be challenging, myocarditis cases could have been missed in these trials [72]. A current report suggests that the prevalence of ICI-associated myocarditis is 1.14% [73]. Under combinational ICI therapy, the incidence rises to 2.4% [73]. However, the true incidence may be even higher due to multiple factors, like the lack of conventional clinical symptoms, challenges in making the diagnosis, and a general lack of awareness of this condition. 

A well-established risk factor for ICI-associated myocarditis is the receipt of combinational ICI therapy. Combination of nivolumab and ipilimumab is accompanied by a 4.74-fold increased risk for myocarditis compared to nivolumab monotherapy [64]. Moreover, myocarditis resulting from combination therapy is associated with a more severe presentation and a higher fatality rate [74]. In addition, studies indicate that alterations in the immune system itself may be a risk factor for ICI myocarditis. It has been shown that clonal cytotoxic Temra CD8+ cells (a subset of T-cells) are significantly increased in the blood of patients with ICI myocarditis [75]. Temra cells are terminally differentiated effector memory cells re-expressing CD45RA and associated with strong effector activity such as killing and cytokine release but a low proliferation rate [76]. Interestingly, an observational study has identified a lower rate of ICI myocarditis in patients with influenza vaccination [77].

Pre-existing cardiovascular risk factors may also be associated with the development of ICI-associated myocarditis. In a multicenter registry, patients who develop myocarditis exhibited a higher prevalence of hypertension and smoking. However, it is important to note that this interpretation is based on univariate analysis rather than multivariate regression analysis [78].

#### 5.1.2. Clinical Presentation

In most cases, ICI myocarditis occurs soon after the initiation of ICI therapy, with the majority occurring within 3 months [73]. However, there is a wide variability in the reported time to onset of symptoms after ICI treatment. In a cohort reported by Escudier, the time to myocarditis diagnosis ranged from 2 days to 454 days, with a median of 65 days [79]. This wide range has also been observed by Moslehi et al., who describe the diagnosis of myocarditis between days 5 and 155, with the median on day 27 [74]. Clinical awareness should start right after the first doses of ICI therapy and continue even if the patient has been on long-term ICI therapy. 

ICI-associated myocarditis can present with a wide spectrum of symptoms of varying severity. From chest pain, shortness of breath, and fatigue to fulminant presentations such as hemodynamic instability, life-threatening arrhythmias, multiorgan failure, and sudden death, the manifestation of ICI myocarditis is highly variable [62,64,73,79]. Fulminant cases with high mortality are characterized by early onset and concomitant myositis and myasthenia gravis [9,74]. At the same time, it can also present as an asymptomatic elevation of cardiac biomarkers, which is referred to as “smoldering myocarditis” [80]. 

#### 5.1.3. Diagnosis

Due to its variable presentation, the diagnosis of ICI-associated myocarditis can be challenging. A high degree of suspicion is required, especially since it can progress rapidly and lead to fatal outcomes. Multiple tests are required to exclude alternative diagnoses, such as acute coronary syndrome, Tako-Tsubo cardiomyopathy, viral myocarditis, or pneumonitis. The gold standard for the diagnosis of myocarditis is histopathological evidence on myocardial biopsy or autopsy [81]. However, a biopsy can be challenging to obtain, and false negatives may occur from sampling error [82]. Therefore, a multipronged approach has been proposed, which includes a combination of ECG, biomarker tests, cardiac imaging, and biopsy [83,84,85]. 

According to the 2022 ESC Guidelines on cardio-oncology, ICI myocarditis can be diagnosed pathohistologically or clinically. Pathohistological diagnosis requires proof of multifocal inflammatory cell infiltrates with overt cardiomyocyte loss in endomyocardial biopsy samples. Clinical diagnosis is based on troponin elevation with one major criterion or two minor criteria after the exclusion of acute coronary syndrome and infectious myocarditis. The major criterion is a positive cardiac MRI result that is diagnostic for acute myocarditis. Minor criteria include typical clinical syndrome, ventricular arrhythmia and/or new conduction system disease, a decline in LV systolic function, concomitant other immune-related adverse events, and a suggestive cardiac MRI [69]. Figure 3 summarizes the diagnosis and management of patients with ICI-associated myocarditis. 

#### 5.1.4. Pathomechanism

The heart is an immune-privileged organ, and immune responses in the heart are particularly dangerous as they can lead to fatal arrhythmias and severe heart failure. Due to its non-regenerative character, susceptibility to arrhythmias even with small damage, and dense vascularity, the heart is prone and vulnerable to immune damage. Therefore, different mechanisms exist to dampen immune responses in the heart [86]. First, under baseline conditions, the heart contains relatively few T-cells. Macrophages and dendritic cells dominate the immune landscape in the myocardium and perform a regenerative function [87]. Second, both central and peripheral tolerance mechanisms limit T-cells directed against myocardial antigens. Third, T-cell-mediated injury to the heart is controlled by multiple negative feedback loops. Secretion of IFN-γ by Th-1 cells and CD8+ cytotoxic T-lymphocytes leads to upregulation of PD-L1 in cardiac endothelial cells, which in turn suppresses effector T-cells [88]. In addition, IFN-γ induces differentiation of monocyte-derived dendritic cells, resulting in abundant nitric oxide production, which blocks differentiation and expansion of Th1 and Th17 cells [89]. 

In myocarditis patients, cardiac-specific anti-myosin autoantibodies and cardiac antigen-specific T-cells have been identified [90,91]. Furthermore, impaired negative selection of CD4^+^ T-cells specific for the alpha-myosin heavy chain in the thymus of mice and humans has led to the development of myocarditis [92]. Indeed, histopathological samples from patients with ICI myocarditis revealed increased myocardial infiltration of T-lymphocytes (both CD4 and CD8) and macrophages. B-lymphocytes have not been noted [64,93]. The importance of T-cells and the immune checkpoint system has been confirmed in preclinical mouse studies. However, there is currently not sufficient data regarding the explicit roles of CD4 or CD8 T-cells in ICI-induced myocarditis. A proficient statement is, at the current state of research, challenging. 

Genetic knock-out of Ctla4 in mice leads to lymphoproliferative disorders, multiorgan immune infiltration, tissue inflammation, including myocarditis, and premature mortality [94,95]. Antibody-mediated CTLA-4 blockage enhances the severity of experimental autoimmune myocarditis [96]. Interestingly, Treg-specific deletion of Ctla4 was sufficient to induce myocarditis development [94,95]. The importance of CTLA-4 for the T-cell population during myocarditis was further highlighted in an induced-myocarditis model. Injection of T-cells deficient in Ctla4 triggered a more severe myocarditis compared to animals receiving Ctla4-positive T-cells [97]. These results point to the crucial role of CTLA-4 for immune homeostasis in the heart. 

In contrast, disruption of the PD-1/PD-L1 axis led to variable phenotypes in mice. Knock-out of Pdcd1 (encoding PD-1) in BALB/c mice leads to increased anti-troponin-I antibodies and dilated cardiomyopathy rather than myocarditis [98,99]. However, germline deletion of Pdcd1 in C57BL/6 did not elicit a cardiac phenotype [100]. This was also observed in C57BL/6 mice with deletions of PD-L1 and PD-L2 [101]. To further complicate the role of PD-1, an independently generated Pdcd1 knock-out on a BALB/c background demonstrated no cardiac abnormalities [102]. However, when this model was challenged with an immune stimulus, it developed overt myocardial inflammation with increased T-cell infiltration [102]. These discrepancies highlight the complex and subtle role of the PD-1/PD-L1 axis and the influence of genetic background and possibly environmental factors. 

In an autoimmune-prone background (such as the MRL-lpr −/−, lacking FAS and predisposing to the development of a systemic lupus erythematosus-like phenotype), genetic or pharmacological disruption of the PD-1 axis results in autoimmune myocarditis and T-cell infiltration [103,104]. This phenotype was also observed when PD-1 and LAG-3 were concomitantly absent [105]. Similarly, monoallelic deletion of Ctla4 with homozygous loss of Pdcd1 leads to myocardial infiltration by T-cells and macrophages, cardiac inflammation, and premature death [106]. These results suggest that compensation mechanisms exist for PD-1 loss, but additional interference in the immune regulation system can lead to a loss of immune homeostasis. Myocarditis develops in a gene–dose-dependent manner.

As described above, loss of PD-1 can lead to cardiac damage. At the same time, the PD-1 axis is actively involved during the cardiac response to injury. In cardiac samples from patients with ICI myocarditis, high levels of membrane and cytoplasmic PD-L1 expression have been detected [64,107]. In mice, PD-L1 is upregulated during myocardial injury induced by cytotoxic T-lymphocyte-mediated myocarditis [88]. Similar upregulation of PD-L1 has been observed in hearts following an ischemic injury [108,109]. 

As PD-L1 expression is upregulated by interferon-γ [88], it has been suggested that PD-L1 upregulation in the myocardium is probably a cytokine-mediated cardioprotective mechanism to limit T-cell-mediated inflammation in states of cardiac stress and disease. ICI application abrogates this protective mechanism of the heart. 

Another proposed mechanism is the clonal expansion of T-cells targeting a homologous antigen shared by the tumor and myocardium. In two patients with ICI-associated myocarditis and myositis, similar T-cell clones were found in the myocardium, skeletal muscle, and tumor [64]. However, the antigen targets of these T-cell clones remain unidentified. Different hypotheses have been postulated. First, striated muscle peptides within cancer tissue elicit a T-cell response, which secondarily damages cardiac and skeletal muscle. Another possibility is that cardiac cells express a relevant cancer antigen. A third possibility is molecular mimicry, in which a cancer antigen bears structural similarity with a cardiac antigen. Further research is needed to identify the exact antigens that elicit T-cell responses during ICI-associated myocarditis. 

Recently, a new mechanism has come to attention. Multi-omic analysis of blood samples from patients with ICI-associated myocarditis showed increased proportions of clonal cytotoxic Temra CD8+ cells [75]. Temra cells are effector memory T cells that re-express CD45RA (a marker on naïve T cells) and are characterized by a highly cytotoxic phenotype [110]. Progressive loss of T-cell function, which is termed T-cell exhaustion, is commonly seen in cancer [111]. ICI treatment reinvigorates exhausted T-cells to induce cytotoxicity against tumor tissue [112]. Overt cytotoxicity of T cells could be a cause of myocarditis development. 

#### 5.1.5. Treatment

The current treatment of ICI-associated myocarditis focuses mainly on immunosuppression. According to the European Society of Cardiology and the American Society of Clinical Oncology, initial management consists of immediate cessation of ICI and the application of glucocorticoids [69,113]. If no significant improvement or clinical deterioration occurs, steroid-resistant ICI-associated myocarditis is confirmed, and second-line immunosuppression should be considered [114,115,116]. These include mycophenolate mofetil, anti-thymocyte globulin (anti-CD3 antibody), i.v. immunoglobulin, plasma exchange, tocilizumab, abatacept (CTLA-4 agonist), alemtuzumab (anti-CD52 antibody), and tofacitinib. Data regarding these agents are so far limited to case series reports and prospective clinical trials that are currently ongoing. 

### 5.2. Non-Inflammatory Cardiomyopathy

#### 5.2.1. Dilated Cardiomyopathy

With increased awareness of cardiac irAEs and improved surveillance strategies, more subtle and comprehensive forms have been detected. On the one hand, “smoldering myocarditis” is characterized by elevated troponin with no cardiac symptoms; on the other hand, dilated cardiomyopathy develops without association with ICI myocarditis. These patients depict signs of left ventricular dysfunction without evidence of inflammation in cardiac PET/MRI or endomyocardial biopsy [117,118]. Similar observations have been made in preclinical mouse models. BALB/c mice with Pdcd1 ablation develop dilated cardiomyopathy with the deposition of IgG antibodies on the surface of cardiomyocytes [98,99]. Furthermore, in a melanoma mouse model, treatment with ICIs leads to a moderate decrease in LVEF, which exacerbates after inotropic stress [119]. A slight increase in myocardial lymphocytes was also observed but remained below the cut-off level for myocarditis [119]. In a small patient cohort of melanoma patients, initiation of ICIs leads to a reduction in LVEF without signs of myocarditis [119]. The exact mechanism remains unclear, and further research is required.

#### 5.2.2. Tako-Tsubo Cardiomyopathy

Tako-Tsubo cardiomyopathy or syndrome (TTS) is a non-inflammatory cardiomyopathy that is also known as stress-induced cardiomyopathy or broken heart syndrome. It is characterized by a transient regional wall motion abnormality, including left ventricular apical ballooning. Clinical symptoms resemble an acute coronary syndrome with chest pain, elevated cardiac enzymes, and ECG changes [120,121] 

There have been several case reports of TTS following treatment with ICIs [122,123]. In the WHO VigiBase pharmacovigilance database study, 13 cases of TTS have been noted [9]. However, TTS remains a diagnosis of exclusion. Acute coronary syndrome and ICI-associated myocarditis must be ruled out. The mechanism of ICI-associated Tako-Tsubo cardiomyopathy is currently unknown. ICIs may trigger a sudden release of large quantities of epinephrine, leading to cardiac stunning. A different hypothesis suggests a direct effect of ICIs on the coronary vasculature, causing multivessel coronary spasm [124]. As ICI-associated TTS is relatively uncommon, research has been rare, and the underlying mechanism remains to be elucidated. 

#### 5.2.3. Ischemic Cardiomyopathy

Ischemic cardiomyopathy is a highly prevalent cardiomyopathy caused by insufficient blood supply to the heart. The underlying pathological mechanism is vasculoproliferative diseases, such as atherosclerosis or myointimal hyperplasia, which lead to the narrowing of the coronary arteries and the development of coronary artery disease and myocardial infarction [125,126]. 

Recently, an increased incidence of myocardial infarction has been observed in patients under ICI therapy. In a matched cohort study including more than 5500 patients, the hazard ratio for a myocardial infarction was about 7-fold in the ICI group [10]. A recent meta-analysis confirmed the increased risk of myocardial infarction under ICI therapy [68]. As expected, the survival of patients with vascular events was worse than those without events [127]. 

Vascular inflammation is a key component of atherosclerosis and leads to plaque formation, progression, and rupture [128]. While vascular plaques cause luminal narrowing and reduced blood supply, rupture can lead to acute vessel occlusion and organ infarction. Since plaque T-cells express high levels of PD-1, intervention in the immune checkpoint system could influence plaque homeostasis [129]. Indeed, animal studies demonstrated increased plaque formation after the blockade of CTLA-4 or PD-1 signaling [130,131,132,133,134]. Similar observations were made in human studies. Calabretta et al. analyzed vascular inflammation via ^18^F-FDG uptake in patients treated with ICIs and found that ICI therapy significantly increased atherosclerotic inflammatory activity [135]. At the same time, Drobni et al. detected an increased rate of progression of atherosclerotic plaque volume after ICI therapy [10]. In contrast to the aforementioned studies, a recent study by Poels et al. did not find a significant change in vascular inflammation after 6 weeks of ICI therapy [133]. A matched autopsy study compared the atherosclerotic plaques of ICI-treated patients and matched controls [136]. This study did not find significant differences in the grade of atherosclerosis or the plaque/median area ratio. However, the immune cell composition (CD3+ T cells/CD68+ macrophages) in the plaque changed after ICI therapy [136]. 

Despite having not fully unveiled the exact pathomechanism between ICI and atherosclerosis/myocardial infarction and subsequent ischemic cardiomyopathy, the high incidence of myocardial infarction in patients receiving ICIs should raise a high level of clinical suspicion. CV risk factors should be periodically screened and adequately treated according to the latest clinical practice guidelines. Patients with symptoms and electrocardiogram findings consistent with myocardial infarction should be treated with standard of care and undergo coronary angiography with possible percutaneous coronary intervention, if needed.

## 6. Conclusions

ICI-mediated cardiotoxicity is an emerging new clinical entity that continues to expand as more patients receive ICIs and have prolonged survival. Besides ICI myocarditis, non-inflammatory cardiomyopathies like Tako-Tsubo cardiomyopathy, dilated cardiomyopathy, ischemic cardiomyopathy, or myocardial infarction start to come to the fore. Despite being relatively rare, the high mortality rate of these complications could affect the further deployment of ICI therapy and urge the need for better strategies to overcome those serious cardiac side effects. Therefore, more studies are required to determine the precise frequency of ICI-mediated cardiotoxicity. By extending clinical data surveillance, diagnostic and therapeutic strategies can be sharpened [137]. Recently, the ESC published the first cardio-oncology guideline, including the management of patients under ICI treatment. However, the majority of recommendations are based on the consensus opinion of experts, small studies, retrospective studies, or registries (Level of Evidence C) [69]. Large clinical trials are required to consolidate the current recommendations. In addition, further investigation of the pathogenesis of ICI-associated cardiomyopathies at the molecular and cellular level is crucial, as this sets the foundation for developing new diagnostic and therapeutic strategies. At the same time, it is of paramount importance for healthcare professionals to have a good understanding of this increasingly common clinical problem. Close collaborations among cardiologists, oncologists, radiologists, and immunologists in both clinical practice and basic science will lead to an improved understanding of ICI-associated cardiomyopathies and eventually a decrease in the lethal capabilities of ICI-induced cardiotoxicity.

## Figures and Tables

**Figure 1 biology-12-00472-f001:**
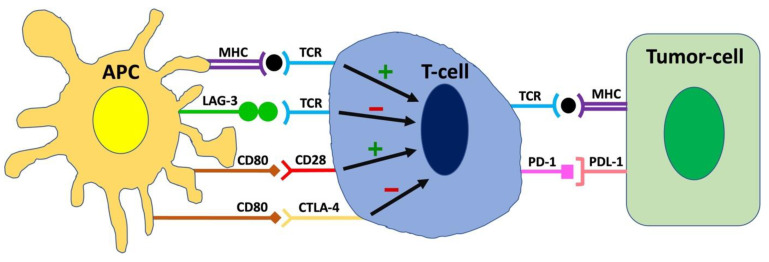
Interaction of immune checkpoints from antigen-presenting cells and tumor-cells with T-cells. Activation and proliferation of T-cells depend on the net results of active stimulatory and inhibitory signals. APC = antigen-presenting cell; CD = cluster of differentiation; CTLA-4 = cytotoxic T-lymphocyte antigen 4; MHC = major histocompatibility complex; PD-1 = programmed cell death 1; PD-L1 = programmed cell death ligand 1; TCR = T-cell receptor; LAG-3 = lymphocyte activation gene-3.

**Figure 2 biology-12-00472-f002:**
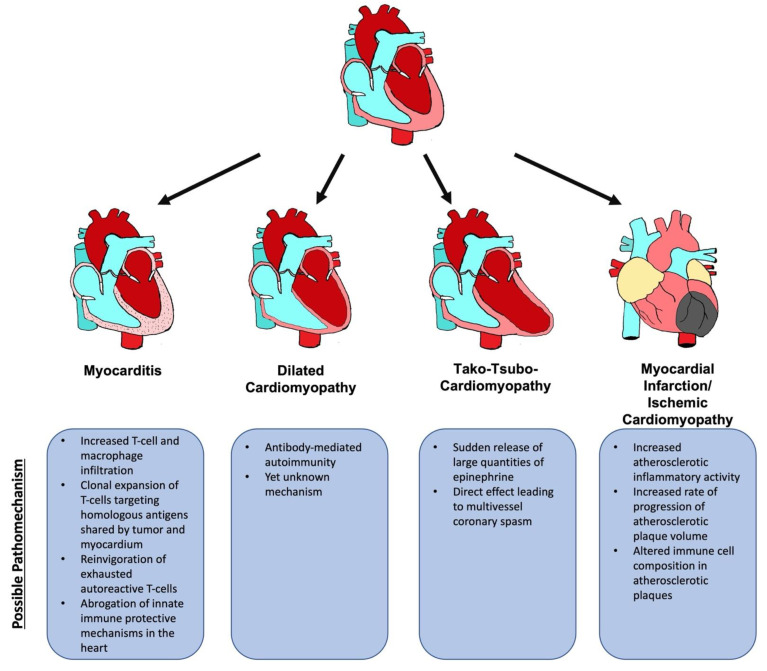
ICI-induced cardiomyopathies and possible underlying pathomechanisms. Besides myocarditis, non-inflammatory cardiomyopathies, such as dilated cardiomyopathy, Tako-Tsubo-cardiomyopathy, myocardial infarction, and ischemic cardiomyopathy, have been observed under ICI therapy.

**Figure 3 biology-12-00472-f003:**
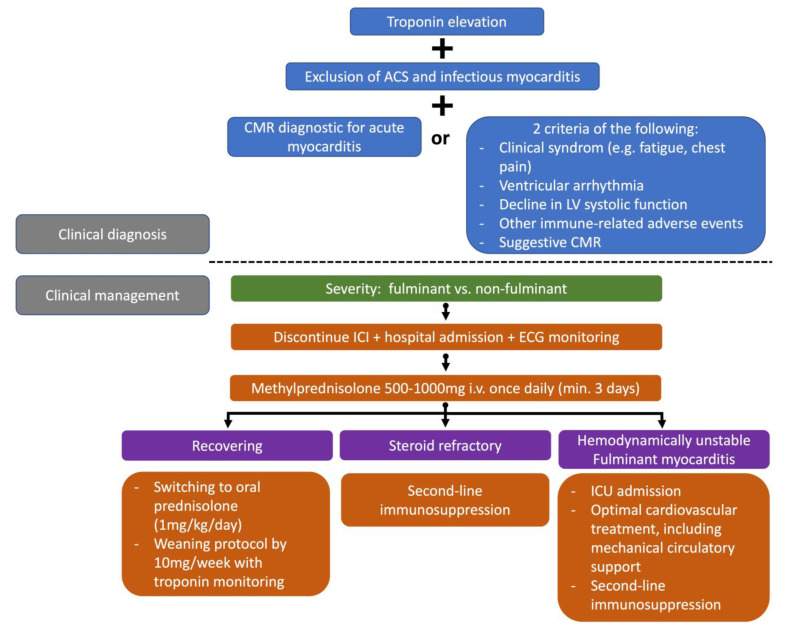
Clinical diagnosis and management of ICI-induced myocarditis (based on the ESC guidelines on cardio-oncology 2022). ACS, acute coronary syndrome; CMR, cardiac MRI; LV, left ventricular; ICU, intensive care unit.

**Table 1 biology-12-00472-t001:** FDA-approved immune checkpoint inhibitors with approved indications (as of February 2023).

Drug	Target	Approval	Indications
Ipilimumab (Yervoy^®^)	CTLA-4	2011	Melanoma, renal cell carcinoma, colorectal cancer, hepatocellular carcinoma, non-small cell lung cancer, malignant pleural mesothelioma, esophageal cancer
Nivolumab (Opdivo^®^)	PD-1	2014	Melanoma, non-small cell lung cancer, malignant pleural mesothelioma, renal cell carcinoma, classical Hodgkin lymphoma, squamous cell carcinoma of the head and neck, urothelial carcinoma, colorectal cancer, hepatocellular carcinoma, esophageal cancer, gastric cancer, gastroesophageal junction cancer, esophageal adenocarcinoma
Pembrolizumab (Keytruda^®^)	PD-1	2014	Melanoma, non⁠-small cell lung cancer, head and neck squamous cell carcinoma, classical Hodgkin lymphoma, primary mediastinal large B-cell lymphoma, urothelial carcinoma, non-muscle invasive bladder cancer, colorectal cancer, gastric cancer, esophageal cancer, cervical cancer, hepatocellular carcinoma, Merkel cell carcinoma, renal cell carcinoma, endometrial carcinoma, cutaneous squamous cell carcinoma, triple-negative breast cancer
Atezolizumab (Tecentriq^®^)	PD-L1	2016	Non-small cell lung cancer, small cell lung cancer, hepatocellular carcinoma, melanoma, alveolar soft part sarcoma
Durvalumab (Imfinzi^®^)	PD-L1	2017	Non-small cell lung cancer, small cell lung cancer, biliary tract cancer, hepatocellular carcinoma
Avelumab (Bavencio^®^)	PD-L1	2017	Merkel cell carcinoma, urothelial carcinoma, renal cell carcinoma
Cemiplimab (Libtayo^®^)	PD-1	2019	Cutaneous squamous cell carcinoma, basal cell carcinoma, non-small cell lung cancer
Dostarlimab (Jemperli^®^)	PD-1	2021	Endometrial cancer
Relatlimab (Opdualag^®^, combination with Nivolumab)	LAG-3	2022	Melanoma

**Table 2 biology-12-00472-t002:** Clinical symptoms, diagnostic and treatment management of ICI-associated myocarditis and cardiomyopathies [69,70,71].

	Symptoms	Diagnosis	Treatment
Myocarditis	Dyspnea Chest pain Fatigue Palpitations Decreased exercise tolerance Syncope	Troponin Echocardiography ECG Cardio-MRI PET (if CMR is not available or contraindicated) EMB (if suspected, but not confirmed by non-invasive diagnostic)	High dose methylprednisolone If steroid refractory, switch to second-line immunosuppression Cessation of ICI ECG monitoring Complications should be treated as per specific ESC Guidelines
Dilated cardiomyopathy	Dyspnea/ orthopnea Edema Fatigue	Echocardiography ECG Exclusion of myocarditis, TTC, ACS	No indication for immunosuppression Treatment according to ESC guidelines for heart failure Interruption of ICI depending on multidisciplinary decision
Tako-Tsubo cardiomyopathy	Chest pain Shortness of breath Palpitations Syncope	Echocardiography Troponin, CK, CK-MB ECG Coronary angiography (exclusion of ACS)	No indication for immunosuppression Treatment according to ESC guidelines for heart failure Interruption of ICI depending on multidisciplinary decision
Myocardial infarction	Chest pain Shortness of breath Palpitations Syncope	Troponin, CK, CK-MB ECG Echocardiography Coronary angiography	No indication for immunosuppression Treatment according to ESC guidelines for acute coronary syndrome Interruption of ICI depending on multidisciplinary decision

## Data Availability

Not applicable.

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
