# Peer review of "Immune Checkpoint Inhibitor Associated Myocarditis and Cardiomyopathy: A Translational Review"

_biology, 2023, doi:10.3390/biology12030472_

Round 1
Reviewer 1 Report
Major comments:
Page 2, Line 60 – mechanism of CD4 versus CD 8 lineage choice needs to be written.
Page 3, Line 118 – The author needs to mention PD–1 suppresses T cell activation through cell-intrinsic mechanisms.
Page 3, Line 130 - Immune checkpoint inhibitors as cancer therapy
Page 5, Line 200 - Pathogenesis of ICI-associated myocarditis at the molecular and cellular levels needs to be highlighted.
Author Response
We thank reviewer 1 for his comments. However due to the shortness of his comments and incongruency to the line reference, it is challenging to fully understand his suggestions. We addressed his comments attached in a point-to-point response:
- We included a description of CD4 vs. CD8 T cells in section 2. Both T-cell subsets express immune checkpoints, which are regulators of the activity process (descriped in detail in Section 2). ICI are unspecific and affects both CD4 and CD8 activity. However, there is currently not sufficient data regarding the explicit roles of CD4 or CD8 T-cells in ICI induced cardiomyopathies. A proficient statement is at the current state of research challenging.
- On page 3 line 106 we state that PD-1 inhibits T-cell activation through a cell-intrinsic signaling mechanism.
- We thank the reviewer for this comment. The current medical application of ICI as cancer therapy with the current indications have been described in detail in Section 3 and Table 1.
- We fully agree with the reviewer, that the pathomechanisms of ICI-associated myocarditis at the molecular and cellular level needs to be highlighted. For this reason, we have written a separate section in this review on this issue (Section 5.1.4.)

Reviewer 2 Report
In this manuscript, Wang et al. reviewed" immune checkpoint inhibitor associated myocarditis and cardiomyopathy". The review is very interesting and provides useful information about a current need in medicine. I would suggest to make the following corrections/additions to improve the quality of this review before the manuscript is suitable for publication.
Major comments
1. Table 1 plus "indications" or" approved use".
2. The content of figure 2 could be more enriched. It must be able to illustrate the possible mechanisms of various cardiac injuries caused by changes in the immune system after ICIs treatment.
3. It is recommended to add table 2 to present in clinical presentation, diagnosis, and ICI treatment for different cardiomyopathies; please refer to C Zito et al. Cancers 2022, 14, 5403.
In this way, it can increase the article's readability.
Minor comments
There are many symbol errors in the use of "," and"." in the text, such as page 4, line 167:5,6%, line 170: 32.518 patients.
Author Response
We appreciate the valuable input from reviewer 2 and have addressed his concerns and adapted the manuscript. Attached is a point-to-point response to the reviewer:
- We have added the current indications to Table 1.
- We thank the reviewer for this suggestion. We modified Figure 2 and added the possible mechanisms behind the various cardiac injuries.
- Table 2 has been added to the manuscript, in which the clinical symptoms, diagnostic and treatment management of the various ICI-associated cardiomyopathies are described.

Round 2
Reviewer 1 Report
This review article by Dong Wang et al provides a compressive review of the current understanding of immune checkpoint inhibitor-associated myocarditis and cardiomyopathy. The authors present a thorough analysis of the potential mechanisms underlying these adverse events and highlight the importance of early recognition and management.
Major Comments:
1) The review article could benefit from a more detailed incidence and prevalence of immune checkpoint inhibitor-associated myocarditis and cardiomyopathy. This could help readers better understand the scope of the problem and the need for increased vigilance and awareness among clinicians.
2) The review article should include potential diagnostic criteria and algorithms for identifying these adverse events, as well as the current best practices for managing these conditions.
3) The review article could impact the development and use of immune checkpoint inhibitors in other types of cancer, as well as the potential need for more targeted and personalized approaches to cancer immunotherapy.
Author Response
- We thank the reviewer for this suggestion. We have included more details to Section 5 about the incidence and prevalence of ICI associated cardiomyopathies. The incidence of ICI-associated myocarditis has been described in Section 5.1.
- A detailed step by step diagnosis and management scheme was added to the manuscript (Figure 3) for ICI-induced myocarditis. For the diagnosis and management of other ICI-induced cardiomyopathies there is currently no specific guidelines. The ESC recommends following the 2021 ESC Guidelines for acute and chronic heart failure.
- We fully agree with the reviewer. With the development of new ICI’s, their approval for new indications and their adoption by healthcare systems, more patients will become eligible for this treatment. Despite being relatively rare, ICI-associated cardiotoxicity are serious and are life-threatening complications. Thus, a close collaboration between the oncologist and cardiologist is of paramount importance to enable the prompt recognition and management of ICI-related CV adverse events. We have included a statement in the conclusion section to highlight the importance for healthcare professionals to recognize this growing clinical entity. At the same time, we believe that progress in research will lead to more personalized cancer therapeutics with fewer side effects and will improve the treatment of cancer patients.
Reviewer 2 Report
Accept in present form
Author Response
We thank the reviewer for his/her appreciation。
Round 3
Reviewer 1 Report
Thanks for answering all the comments.